# Boosting Prompt-Based Self-Training With Mapping-Free Automatic Verbalizer for Multi-Class Classification

**Yookyung Kho, Jaehee Kim, Pilsung Kang**

Korea University, Seoul, Republic of Korea

{yookyung_kho, jaehee_kim, pilsung_kang}@korea.ac.kr

## Abstract

Recently, prompt-based fine-tuning has garnered considerable interest as a core technique for few-shot text classification task. This approach reformulates the fine-tuning objective to align with the Masked Language Modeling (MLM) objective. Leveraging unlabeled data, prompt-based self-training has shown greater effectiveness in binary and three-class classification. However, prompt-based self-training for multi-class classification has not been adequately investigated, despite its significant applicability to real-world scenarios. Moreover, extending current methods to multi-class classification suffers from the verbalizer that extracts the predicted value of manually pre-defined single label word for each class from MLM predictions. Consequently, we introduce a novel, efficient verbalizer structure, named Mapping-free Automatic Verbalizer (MAV). Comprising two fully connected layers, MAV serves as a trainable verbalizer that automatically extracts the requisite word features for classification by capitalizing on all available information from MLM predictions. Experimental results on five multi-class classification datasets indicate MAV's superior self-training efficacy.[1]

## 1 Introduction

The language model has demonstrated impressive results in numerous practical applications by undergoing extensive pre-training using objectives such as masked language modeling and autoregressive language modeling (Peters et al., 2018; Radford and Narasimhan, 2018; Devlin et al., 2019). Recently, the prompting approach, as proposed by Radford et al. (2019) and Brown et al. (2020), has played a critical role in addressing the few-shot scenario in various natural language processing tasks (Chen et al., 2022; Ma et al., 2022; Liu et al., 2023). Schick and Schütze (2021) and Gao et al. (2021a) introduced a novel prompt-based fine-tuning methodology, known as PET[2] and LM-BFF[3], which applied prompting to encoder-only models such as BERT (Devlin et al., 2019) and RoBERTa (Liu et al., 2019). Their research demonstrated the efficacy of prompt-based fine-tuning in few-shot text classification tasks, as it significantly outperformed standard fine-tuning methods.

To address the scarcity of labeled data, semi-supervised learning approaches have also been extensively researched by leveraging unlabeled data, which often contain sufficient contextual information (Xie et al., 2020; Chen et al., 2020; Li et al., 2021). Recent studies (Schick and Schütze, 2021; Chen et al., 2021; Zhao and Yao, 2022; Wang et al., 2022b) have experimentally demonstrated the efficacy of combining prompt-based fine-tuning with self-training. However, existing research on prompt-based self-training has focused on addressing binary classification (e.g., Sentiment Analysis) or three-class classification (e.g., Natural Language Inference) tasks (Chen et al., 2021; Wang et al., 2022b). Although leveraging unlabeled data has been a crucial aspect of multi-class classification (Song et al., 2011; Li et al., 2019; Tang et al., 2022), prompt-based self-training approaches for multi-class classification remain under-explored.

The challenge in applying conventional prompt-based self-training approaches to multi-class classification arises from the verbalizer. In prompt-based fine-tuning, a given input sentence **x** (e.g., "I am happy.") is wrapped with a pre-defined natural language sequence including [MASK] token, called *template* (e.g., "**x**: It is [MASK]."). Subsequently, *verbalizer*, a mapping from a label word (e.g., "good") to a specific class (e.g., positive), extracts predicted value of the label word from the MLM prediction to generate the final predictive probability distribution. That is, the probability

---

[1] Our code is publicly available at https://github.com/yookyungkho/MAV.

[2] Pattern Exploiting Training
[3] Better Few-shot Fine-tuning of Language Models

that the label word is predicted at the masked position is regarded as the prediction probability of the class.

The main problem is that most of self-training studies construct verbalizers using manually selected single label words. Creating such a manual verbalizer necessitates domain knowledge (Gao et al., 2021a), which causes the high cost associated with manually selecting label words for numerous classes under the constraint of choosing a label word as a single token. Furthermore, extracting only a small amount of limited information corresponding to label words from high-dimensional MLM predictions can result in information loss (Hu et al., 2022), leading to considerable performance disparities depending on the selected label word (Gao et al., 2021a; Webson and Pavlick, 2022).

To address the limitations of current verbalizers, we propose a novel Mapping-free Automatic Verbalizer (MAV) that consists of two fully connected layers (FCLs) to automatically extract the necessary vocabulary information for classification by leveraging entire MLM predictions without specifying explicit label words. The proposed methodology MAV has the following advantages: (1) It eliminates the need for manual selection of label words. (2) It leverages the entire MLM predictions without information loss. (3) It circumvents the cost issue associated with searching for optimal label words. (4) It can be applied to any multi-class datasets, irrespective of the number of classes.

In this study, we conducted few-shot text classification on five multi-class datasets, achieving an average performance improvement of 12.8% over existing self-training methodology. Quantitative metrics demonstrated that the proposed MAV benefits most from self-training. Further analysis revealed that MAV is proficient in extracting vocabulary features representative of each class, thereby enhancing the efficacy of prompt-based learning without manual manipulation of the verbalizer.

## 2 Related Work

### 2.1 Prompt-based fine-tuning

Prompt-based fine-tuning reformulates the fine-tuning objective to MLM objective. This approach ensures that pre-trained knowledge is fully leveraged in the fine-tuning process (Gao et al., 2021a; Han et al., 2022; Liu et al., 2023), while also incorporating task-specific information through prompts (Schick and Schütze, 2021). Various studies have aimed to answer to a critical research question regarding verbalizer, which significantly influences prompt-based learning performance (Gao et al., 2021a; Webson and Pavlick, 2022): *How to identify suitable label words for a given class?*

**Manual verbalizer** Schick and Schütze (2021) manually chose a single label word for each class. Selecting a single label word can encompass various heuristic methods, including utilizing the class name itself or relying on an expert with specialized knowledge.

**Automatic verbalizer** Gao et al. (2021a) highlighted the sub-optimality of manual approach and proposed an automatic search method to determine the optimal single label word combination by evaluating candidates generated through MLM inference. Shin et al. (2020) and Schick et al. (2020) introduced multiple label words search strategy based on gradient and loss. Wang et al. (2022a) performed MLM inference before fine-tuning and designated the top-k tokens as label words. Additionally, Hu et al. (2022) proposed a hybrid manual and automatic verbalizer that expands the label word space by leveraging external knowledge bases.

**Verbalizer-free methods** The cost of label engineering is a main challenge in prompt-based fine-tuning. Thus, verbalizer-free methodologies emphasize the importance of exploiting the [MASK] representation rather than the MLM prediction itself (Cui et al., 2022; Karimi Mahabadi et al., 2022; Xu et al., 2023). These approaches rely solely on the [MASK] representation and employ distance-based learning to bring instances of the same class closer in the representation space.

### 2.2 Prompt-based self-training

Contrary to widely studied prompt-based fine-tuning, prompt-based self-training has received relatively less attention. Chen et al. (2021) introduced SFLM[4], a method combining manual label words from Gao et al. (2021a) with pseudo-labeling and consistency regularization algorithms from FixMatch (Sohn et al., 2020) for self-training. Zhao and Yao (2022) incorporated the consistency regularization loss term from Xie et al. (2020) into the self-training process of SFLM. Wang et al.

---

[4]Self-training techniques and a data-efficient Few-shot learner of Language Model

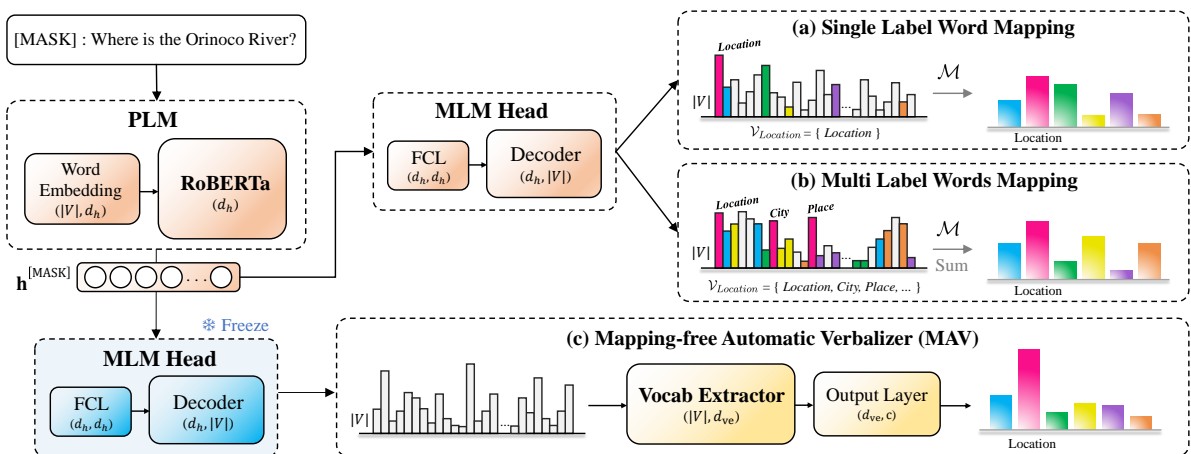

Figure 1: Illustration of MAV and comparison with other verbalizers. (a), (b): The input sequence with the template added is fed into the PLM and MLM Head, which returns a prediction vector of [MASK] token. Among MLM prediction, logit values of one or multiple pre-defined label words of each class are extracted to form the final prediction vector by a mapping function $\mathcal{M}$. (c): MLM prediction is calculated while the parameters of the MLM Head are frozen. MAV feeds MLM prediction into two FCLs to return the final probability distribution, automatically identifying the necessary vocabulary information for class distinction.

(2022b) conducted self-training with a teacher-student-based adapter tuning method. Most of these studies target binary and three-class classification. Furthermore, they all rely on the manual single-label word mapping of Schick and Schütze (2021) and Gao et al. (2021a), potentially causing information loss and sub-optimal performance.

Furthermore, it is challenging to integrate the verbalizer enhancement methodologies mentioned in Section 2.1 with self-training for multi-class classification. Specifically, automatic verbalizer approaches require additional data for label word search and incur a significant computational burden when the number of classes increases. Moreover, verbalizer-free methods rely on spatial distances, making it impossible to generate explicit predictive probability distributions essential for the self-training process. In contrast, our proposed MAV is a specialized verbalization method for multi-class classification that offers cost-saving benefits, as it eliminates the need for label word search. Moreover, MAV utilizes all MLM prediction information, allowing pre-trained knowledge to be fully leveraged in downstream tasks.

## 3 Methodology

### 3.1 Mapping-free automatic verbalizer

The procedure of prompt-based fine-tuning is shown in Figure 1. For a multi-class classification task, the given input sequence $\mathbf{x}$ is augmented with

a pre-defined template as shown in Eq. (1). In this study, we used the manual template ("[MASK] :") from Gao et al. (2021a).

$$\mathbf{x}' = [\text{CLS}] \; [\text{MASK}] : \; \mathbf{x} \; [\text{SEP}]. \qquad (1)$$

The MLM prediction of the [MASK] token $\mathbf{v} \in \mathbb{R}^{|V|}$ is generated by the Pre-trained Language Model (PLM) and the MLM head.

Existing verbalizers (Figure 1 (a) and (b)) construct the final predictive probability distribution with the MLM prediction as

$$P\left(y|\mathbf{x}'\right) = \mathcal{M}\left(P\left([\text{MASK}] = v|\mathbf{x}'\right)|v \in \mathcal{V}_y\right), \quad (2)$$

where $\mathcal{V}_y$ is a set of pre-defined label words of a specific class $y$, and a function $\mathcal{M}$ (e.g., identity function, sum) transforms predicted values of label words to the probability of the class. These approaches require additional costs for label word selection.

In contrast, as shown in Figure 1 (c), Mapping-free automatic verbalizer (MAV) consists of two FCLs and feeds the MLM prediction $\mathbf{v}$ as

$$P\left(y|\mathbf{x}'\right) = \text{softmax}\left(W_c^T \cdot \text{Tanh}\left(W_{\text{ve}}^T \cdot \text{Tanh}\left(\mathbf{v}\right)\right)\right). \qquad (3)$$

$W_{\text{ve}} \in \mathbb{R}^{|V| \times d_{\text{ve}}}$ is a weight matrix of the first FCL, named Vocab Extractor (VE). It extracts useful information from high dimensional ($|V|$) vocabulary features to low dimensional ($d_{\text{ve}}$) representation. $d_{\text{ve}}$ is set to 256 via hyper-parameter tuning

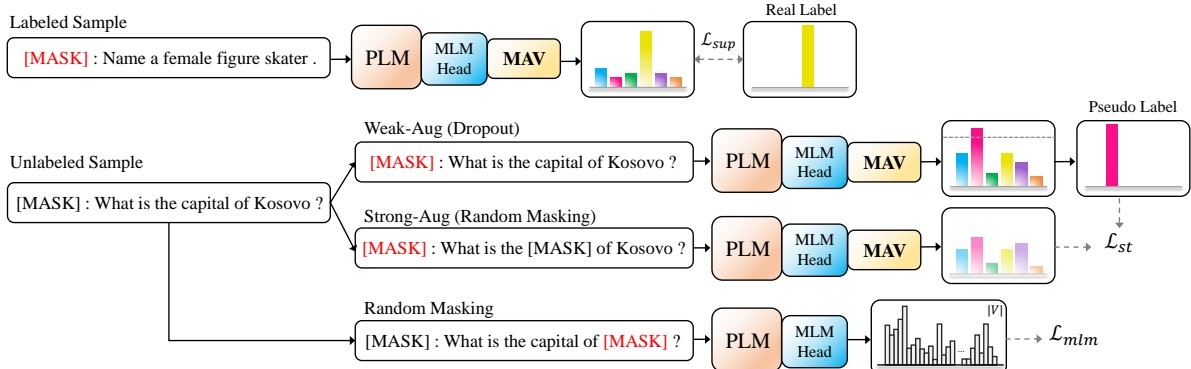

Figure 2: Overall Self-training Framework. The labeled sample with the template is processed through the prompt framework including MAV to return the predictive probability distribution, and the supervised loss is calculated with the ground-truth label. The unlabeled sample is weakly augmented to generate a pseudo-label. The self-training loss is computed with the pseudo label and the predicted probability distribution derived by strongly augmented sample. Furthermore, the auxiliary MLM loss is calculated by applying random masking on the unlabeled sample. Note that [MASK] highlighted in red represents the token used in MLM prediction.

(see Appendix B.2). The subsequent FCL with a weight matrix $W_c \in \mathbb{R}^{d_{ve} \times c}$ generates the final predictive probability distribution for each class based on the compressed information. This approach converts the verbalizer into a learnable form without providing any label word information related to the class. Therefore, MAV autonomously identify the vocabulary information needed to distinguish between classes in the MLM prediction.

Another significant consideration in constructing MAV is the fixed parameters of the MLM head. During the pre-training phase, the decoder of the MLM head shares parameters with the word embedding matrix, meaning it contains pre-trained vocabulary representation information. In order to preserve this information and reduce training cost, the MLM head is frozen during the fine-tuning phase. The effect of freezing parameters is discussed in Section 4.5.

## 3.2 Overall training framework

The overall self-training framework using MAV minimizes supervised and unsupervised losses calculated based on labeled and unlabeled data, as depicted in Figure 2. The unsupervised loss comprises the self-training loss based on Fix-Match (Sohn et al., 2020) and the auxiliary MLM loss, following the previous research, SFLM (Chen et al., 2021). We construct the final loss term with weighting factors, $\lambda_1$ and $\lambda_2$, as

$$\mathcal{L}_{total} = \mathcal{L}_{sup} + \lambda_1 \cdot \mathcal{L}_{st} + \lambda_2 \cdot \mathcal{L}_{mlm}. \quad (4)$$

Let $\mathbf{x}_i$ and $\mathbf{u}_i$ in Eq. (5) denote labeled and unla-

beled samples in the batch, respectively. $B$ is the number of labeled samples in the batch, and $\mu$ is the ratio of labeled and unlabeled samples. Each loss is computed as follows.

$$\begin{aligned} \mathbf{x}_i \ (i \in (1, \cdots, B)), \\ \mathbf{u}_i \ (i \in (1, \cdots, \mu B)). \end{aligned} \quad (5)$$

**Supervised loss**   For a labeled sample $\mathbf{x}_i$, the predictive distribution $\widetilde{\mathbf{y}}_i = P(y|\mathbf{x}_i')$ is obtained from the MAV framework. The Cross Entropy (CE) loss is measured between $\widetilde{\mathbf{y}}_i$ and the ground truth label $\mathbf{y}_i$ expressed as a one-hot vector.

$$\mathcal{L}_{sup} = \frac{1}{B} \sum_{i=1}^{B} \text{CE}(\mathbf{y}_i, \widetilde{\mathbf{y}}_i). \quad (6)$$

**Self-training loss**   $\mathcal{L}_{st}$, a component of the unsupervised loss, aims to perform consistency regularization by taking weak and strong augmentation to the unlabeled sample so that the predictive distribution of the strongly augmented sample closely matches the pseudo-label generated from the weakly augmented sample.

$$\begin{aligned} \mathcal{L}_{st} = \frac{1}{\mu B} \sum_{i=1}^{\mu B} \mathbb{1}\left(max\left(\widetilde{\mathbf{q}}_i^{weak}\right) \geq \tau\right) \cdot \\ \text{CE}\left(\hat{\mathbf{q}}_i^{weak}, \widetilde{\mathbf{q}}_i^{strong}\right), \end{aligned} \quad (7)$$

where $\widetilde{\mathbf{q}}_i^{weak}$ and $\widetilde{\mathbf{q}}_i^{strong}$ denote the predictive distributions of the weakly and strongly augmented samples, respectively. $\hat{\mathbf{q}}_i^{weak}$ represents the pseudo label for the weakly augmented sample, taking

the form of a hard label (one-hot vector). Following the thresholding technique of FixMatch, the self-training loss is computed only for samples where the confidence of the weakly augmented sample $(\max(\widetilde{\mathbf{q}}_i^{weak}))$ exceeds a pre-determined threshold $(\tau)$, set to 0.95. Additionally, we implemented dropout and random masking as weak and strong augmentation methods, following Chen et al. (2021) (see Appendix B.2 for details).

**Auxiliary MLM loss**   $\mathcal{L}_{mlm}$ is an auxiliary loss proposed by Schick and Schütze (2021) to prevent catastrophic forgetting often observed when fine-tuning with a small amount of labeled data. Chen et al. (2021) added a similar self-supervised loss that predicts the [MASK] token of the strongly augmented sample for regularization. In this study, we adopted the auxiliary MLM loss of PET and applied 15% random masking to the unlabeled sample, separate from the strong augmentation.

$$\mathcal{L}_{mlm} = \frac{1}{\mu B} \sum_{i=1}^{\mu B} \mathcal{L}_{mlm}^{i}, \qquad (8)$$

where $\mathcal{L}_{mlm}^{i}$ denotes the average MLM loss of masked tokens (except the one in the template) in the $i^{th}$ sequence:

$$\mathcal{L}_{mlm}^{i} = \frac{1}{m} \sum_{j=1}^{m} -\log P\left([\text{MASK}_j] = t_j \,\big|\, \text{RM}\left(\mathbf{u}_i'\right)\right), \qquad (9)$$

where $m$ represents the number of masked tokens in the $i^{th}$ sequence, and $t_j$ denotes the target of the $j^{th}$ [MASK] token. RM means Random Masking.

## 4 Experiments

### 4.1 Experimental setup

#### 4.1.1 Dataset

We evaluated MAV on five multi-class datasets; TREC (Hovy et al., 2001), TREC50 (Li and Roth, 2002), GoEmotions (Demszky et al., 2020), Yahoo Answers (Zhang et al., 2015), and AG's News (Zhang et al., 2015). The datasets encompass a minimum of four classes and extend up to dozens of classes, thereby confirming versatility across datasets with different numbers of classes.

Following the experimental settings of Chen et al. (2021), we set the number of labeled data per class $(k = 16)$ and unlabeled data ratio $(\mu = 4)$ per class, resulting in a minimum training data size of $80\,(16+16*4)$ for each class. However, for TREC,

| Dataset | Type | # Class (Before) | # Class (After) | $k$ | $\mu$ |
|---|---|---|---|---|---|
| TREC | Topic | 6 | 6 | 12 | 4 |
| Trec50 | Topic | 50 | 22 | 8 | 4 |
| GoEmotions | Emotion | 28 | 26 | 8 | 4 |
| Yahoo Answers | Topic | 10 | 10 | 16 | 4 |
| AG's News | Topic | 4 | 4 | 16 | 4 |

Table 1: Data description. Each column indicates the name of dataset, the type of classification, the number of classes before and after data pre-processing, the number of labeled data per class $(k)$, and the ratio between labeled and unlabeled data size $(\mu)$.

TREC50, and GoEmotioins dataset, some classes did not fulfill the minimum data size per class. In such cases, we maintained a constant ratio of unlabeled data $(\mu)$ but decreased the value of $k$, which means fewer data for each class. Some classes that did not meet the minimum data requirement were excluded from the experiment. Details of datasets are provided in Table 1 and Appendix B.1.

#### 4.1.2 Baselines

Three baselines were employed to evaluate the performance of prompt-based self-training with our proposed MAV.

**Standard fine-tuning** is a traditional fine-tuning technique that does not utilize templates or verbalizers. Instead, it performs classification using the [CLS] token representation by adding a classification head after the PLM.

**Single label word** takes only the logit value of a single label word mapped to a class in MLM prediction. We adopted the manual label words used in Schick and Schütze (2021) and Gao et al. (2021a). Implementation details can be found in Appendix B.3.

**Multi label words** take the logit values of multiple label words mapped to each class in MLM prediction. We adopted AMuLaP[5] (Wang et al., 2022a), a parameter-free automatic verbalizer construction method using only a small number of labeled samples, as the Multi Label Words baseline. Details can be found in Appendix B.3.

Furthermore, we explored the verbalizer-free approach discussed in Section 2 as a baseline. However, this methodology is not applicable to the self-training approach employed in this study. Therefore, we conducted an alternative experiment, which is described in Appendix C.

---

[5] Automatic Multi-Label Prompting

|  |  | TREC | TREC50 | GoEmotions | Yahoo | AG's News |
|---|---|---|---|---|---|---|
| Standard Fine-tuning | Small-supervised | 70.1 (4.6) | 80.0 (6.5) | 21.2 (1.6) | 63.4 (1.8) | 84.6 (0.9) |
|  | Semi-supervised■ | 80.2 (4.2) | 79.4 (9.4) | 21.9 (1.5) | 64.4 (1.4) | 86.3 (1.1) |
|  | Full-supervised | 90.8 (3.1) | 91.2 (2.0) | 36.4 (1.0) | 68.7 (0.3) | 88.5 (0.8) |
|  | Benefit Ratio ↑ | 0.49 | -0.05 | 0.05 | 0.19 | 0.44 |
| Single Label Word | Small-supervised♦ | 77.2 (6.3) | 84.4 (6.0) | 15.7 (2.2) | 65.5 (1.5) | 86.1 (1.0) |
|  | Semi-supervised♣ | 81.4 (3.1) | 81.5 (12.1) | 16.8 (1.0) | 68.0 (0.7) | **87.3 (0.8)** |
|  | Full-supervised | 91.6 (1.0) | 93.1 (1.5) | 32.8 (2.0) | 69.2 (0.9) | 88.1 (0.6) |
|  | Benefit Ratio ↑ | 0.29 | -0.33 | 0.06 | 0.66 | 0.60 |
| Multi Label Words | Small-supervised♠ | 72.7 (6.0) | 64.6 (1.2) | 17.3 (1.6) | 57.6 (2.2) | 81.4 (1.5) |
|  | Semi-supervised | 76.3 (2.3) | 82.0 (2.1) | 18.0 (1.3) | 61.1 (1.7) | 84.1 (0.7) |
|  | Full-supervised | 81.4 (3.1) | 89.7 (3.3) | 32.8 (1.4) | 68.2 (0.5) | 88.1 (0.8) |
|  | Benefit Ratio ↑ | 0.21 | 0.69 | 0.05 | 0.33 | 0.40 |
| MAV (ours) | Small-supervised | 77.1 (9.9) | 78.0 (5.2) | 24.2 (1.7) | 64.7 (2.4) | 84.3 (2.2) |
|  | Semi-supervised | **85.4 (3.7)** | **87.9 (5.1)** | **25.4 (2.5)** | **68.1 (0.9)** | 87.2 (0.9) |
|  | Full-supervised | 93.9 (1.1) | 91.3 (2.0) | 36.1 (1.2) | 69.4 (0.5) | 89.0 (0.5) |
|  | Benefit Ratio ↑ | **0.50** | **0.74** | **0.10** | **0.73** | **0.63** |

Table 2: Results of few-shot multi-class classification. Each cell is filled with the average accuracy of the five random seeds, with numbers in brackets indicating the standard deviation. For each methodology, we report performance of small-supervised, semi-supervised, and full-supervised model, along with a benefit ratio to measure the effectiveness of self-training. Small-supervised and full-supervised models are trained with the supervised loss outlined in Section 3.2, while semi-supervised model is trained with both self-training loss and auxiliary MLM loss in addition to the supervised loss. The highest self-training performance and benefit ratio achieved in each dataset are emphasized in bold text. The superscript symbol denotes previous researh corresponding to each setting (■: FixMatch (Sohn et al., 2020), ♦: LM-BFF (Gao et al., 2021a), ♣: SFLM (Chen et al., 2021), ♠: AMuLaP (Wang et al., 2022a)).

### 4.1.3 Experimental details

**PLM** We adopted RoBERTa-base (Liu et al., 2019) as the PLM for all experiments.

**Evaluation** Following Gao et al. (2021a), we sampled training and validation data under five random seeds and evaluated them using the original test dataset, measuring the average accuracy and standard deviation. To adhere to the few-shot text classification assumption, we set the size of the validation dataset to be the same as that of the training dataset. Furthermore, hyper-parameters are tuned based on validation performance. Details of hyper-parameter selection are in Appendix B.2.

**Quantitative metric** To assess self-training effectiveness, we measured the performance of three settings as follows: **Small-supervised** model trained with a small amount of labeled data ($k$ labeled samples per class), **Semi-supervised** model trained with a small amount of labeled data and $\mu$ times unlabeled data ($k$ labeled and $\mu k$ unlabeld samples per class), and **Full-supervised** model trained with full labeled data ($k + \mu k$ labeled samples per class).

The performances of small-supervised and full-supervised models indicate the lower and upper bounds for the semi-supervised model. In other words, small-supervised performance showcases how semi-supervised model benefits from incorporating unlabeled data with limited labeled data, while full-supervised performance represents the upper limit in the semi-supervised context. We calculated the benefit ratio (Zhu et al., 2022) as

$$Benefit\ Ratio = \frac{Acc\,(Semi) - Acc\,(Small)}{Acc\,(Full) - Acc\,(Small)}, \quad (10)$$

measuring the performance improvement of semi-supervised model compared to small and full supervised model.[6] The closer the benefit ratio is to 1, the more comparable the performance of the semi-supervised model is to the full-supervised model.

### 4.2 Experimental results

Table 2 presents the experimental results for multi-class classification. The proposed MAV achieves

---

[6]The benefit ratio was originally proposed to measure the learning difficulty of each class. In this paper, we aggregated results from all classes to demonstrate the effectiveness of semi-supervised learning.

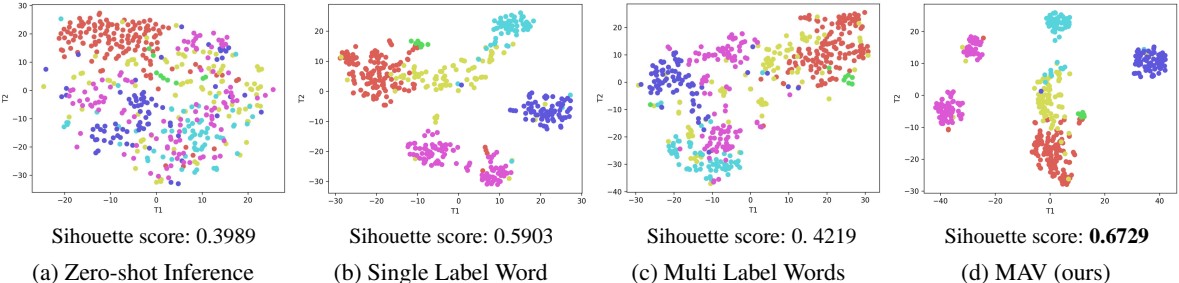

| Sihouette score: 0.3989 | Sihouette score: 0.5903 | Sihouette score: 0. 4219 | Sihouette score: **0.6729** |
|:---:|:---:|:---:|:---:|
| (a) Zero-shot Inference | (b) Single Label Word | (c) Multi Label Words | (d) MAV (ours) |

Figure 3: t-SNE visualization of [MASK] representations from baselines and MAV. (a) Zero-shot inference before fine-tuning, (b) Single Label Word Mapping, (c) Multi Label Words Mapping, (d) Mapping-free Auto Verbalizer results. The numbers below each plot represent Silhouette scores. We used TREC dataset, which contains six classes, and randomly sampled one of the five seeds.

the highest self-training performance across all baselines, except AG's News dataset, with an average self-training performance improvement of 12.8% over the existing prompt-based self-training methodology, SFLM (Single Label Word, Semi-supervised). Moreover, MAV attains the highest benefit ratio for all datasets, demonstrating that it is the most advantageous verbalizer structure for self-training in multi-class classification tasks. In contrast, Single Label Word does not fully benefit from self-training, despite performing well in small-supervised setting. Moreover, Single Label Word requires additional costs to manually specify the label word for every class.

The self-training performance of MAV and Single Label Word baseline is comparable for AG's News, which has fewer classes than other datasets, and for Yahoo Answers, which has a relatively clear distinction between classes. However, MAV offers a clear advantage in that it does not require manual intervention on the label words and can be scaled to other datasets regardless of the number of classes. For the GoEmotions dataset, which involves categorizing over 20 fine-grained emotions, the classification task is challenging, and the impact of self-training is uncertain across all approaches due to a relatively weak classifier. Nonetheless, MAV significantly outperforms Single Label Word and Multi Label Words, which utilize limited information when class distinctions are unclear.

Interestingly, our findings contradict the common belief that prompt-based fine-tuning always outperforms standard fine-tuning in few-shot settings (Gao et al., 2021a; Chen et al., 2021). Our results indicate that standard fine-tuning is not significantly inferior in multi-class classification tasks. Moreover, a considerable drop in performance of Multi Label Words is observed when compared to

standard fine-tuning.

Overall, the proposed MAV maximizes the effectiveness of prompt-based self-training by fully leveraging MLM prediction without any additional cost for label word engineering.

### 4.3 Qualitative analysis

The effectiveness of prompt-based self-training in text classification, which utilizes MLM predictions from [MASK] representations, is heavily influenced by the degree of aggregation of these representations by class (Cui et al., 2022). Therefore, we examined the aggregation of [MASK] representations by class within the test dataset using $t$-SNE plot and Silhouette scores. As shown in Figure 3, MAV (Figure 3 (d)) forms much denser clusters of [MASK] representations compared to zero-shot inference (Figure 3 (a)) and explicit label word mapping methodologies (Figure 3 (b) and (c)). Additionally, we applied K-Means clustering with the number of clusters equal to the number of classes and calculated the silhouette score. Both $t$-SNE plots and Silhouette scores demonstrate that the clustering results of MAV are superior to the baselines. This suggests that leveraging all information from MLM predictions is beneficial for multi-class classification in prompt-based self-training than using limited information from MLM predictions.

### 4.4 Comparison of verbalizers

The core motivation of MAV is to construct a learnable verbalizer that enables the model to identify the vocabulary features helpful for classification. This section investigates whether MAV focuses on the most appropriate vocabularies that represent the unique characteristics of each class. Since the verbalizer of Multi Label Words (Table 3 (b)) was built with only a small amount of labeled data, we

| Class | (a) Single Label Word | (b) Multi Label Words | (c) MAV (ours) |
|---|---|---|---|
| Location | Location* | Poll, Related, RELATED, More, MORE, Video, Query, Bonus, … | Map, city, Map*, Maps, map, Location, cities*, map*, Paris, … |
| Human being | Human* | Next, Note, NOTE, UPDATE, Quick, Advertisement, Edit, Trump, Now, … | Who*, Winners*, Name, winners*, named*, Guests*, nominated*, … |
| Numeric value | Number* | Question, Analysis, Myth, AP, Correction, GM, CNN, BP, Claim, … | 1985, Time, RM, 1968, 1969, 1982, 1965, 1984, Temperature, 1978, … |

Table 3: Comparison of verbalizers. The first column involves three randomly sampled classes in TREC dataset, while each cell of the other three columns indicates (a) a single label word mapped to each class, (b) automatically explored multiple label words through AMuLaP for each class, and (c) vocabulary features that contributed the most to the prediction with each class. We randomly sampled one of the five seeds for (b) and (c). Note that the word with * (e.g., *'Ġacronym'*) refers to a token with a space symbol.

validated the capability of MAV under a more challenging small-supervised setting without the benefit of unlabeled data. SHapley Additive exPlanations (SHAP), proposed by Lundberg and Lee (2017), was utilized to identify the words that contributed the most to the final prediction.

Specifically, we calculated the SHAP value for each class using only the samples that answered correctly in the test dataset and listed top features (tokens), as shown in Table 3 (c). The empirical evidence supports that MAV truly has the ability to utilize words (e.g., *'Map'*, *'city'*, *'Location'*) associated with the class (e.g., *'Location'*) for multi-class classification. This is remarkable since MAV does not refer to any label word information about the class at all. On the other hand, for Multi Label Words, the lowest performer, selecting label words through zero-shot MLM inference with only a small amount of labeled data is unsuitable for multi-class classification tasks where it is relatively difficult to distinguish between classes.

### 4.5 Effect of freezing parameters

An important feature of MAV is that it freezes the parameters of the MLM head that produces vocabulary logits. This freezing mechanism aims to generate informative predictions that are not biased by the small amount of labeled data during the fine-tuning process. Simultaneously, we harnessed the pre-trained knowledge embedded within the decoder of the MLM head. We compared the performance of prompt-based self-training based on the MAV structure with different parameter update scenarios while following the overall self-training framework in Figure 2. According to Table 4, self-training performances of freezing the parameters of the MLM head are almost identical to the results

| Parameter freeze | TREC | TREC50 | GoEmotions |
|---|---|---|---|
| No | **85.7 (4.1)** | 87.4 (2.9) | 25.3 (1.6) |
| MLM Head (ours) | 85.4 (3.7) | **87.9 (5.1)** | **25.4 (2.5)** |
| RoBERTa | 50.1 (8.6) | 36.1 (8.4) | 13.9 (1.7) |

Table 4: Self-training performances of MAV depending on freezing parameters. The three settings are as follows: No (updating all parameters), MLM Head (freezing only the MLM head), and RoBERTa (freezing the RoBERTa encoder).

| | TREC | TREC50 | GoEmotions |
|---|---|---|---|
| MAV (ours) | **83.6 (5.7)** | **87.0 (3.6)** | **25.4 (1.4)** |
| Single Label Word | 82.6 (3.5) | 83.0 (8.2) | 15.2 (0.8) |

Table 5: Self-training performances of FlexMatch.

of the fully fine-tuned case. At the same time, it significantly outperforms the scenario where the RoBERTa encoder is frozen. This suggests that the MLM head is sufficiently adept at generating MLM predictions that carry valuable information about the downstream data even without further tuning.

### 4.6 Effect of self-training methods

We adopted FixMatch as the self-training methodology for a fair comparision with SFLM. To verify the robustness of MAV when implementing different self-training algorithms besides FixMatch, we conducted a comparative analysis using Single Label Word, which demonstrated the best performance among the baseline models. Our experiments included FlexMatch (Zhang et al., 2021), an improved self-training algorithm that adjusts the threshold based on the difficulty of each class. The results are illustrated in Table 5. The performance gaps between MAV and Single Label Word are consistent across the self-training methods.

## 5 Conclusion

In this research, we proposed a simple and effective verbalizing method, Mapping-free Automatic Verbalizer, to enhance prompt-based self-training performance on various multi-class classification tasks. Unlike previous verbalizers which take limited information from MLM prediction and require label engineering, MAV is a trainable verbalizer consisting of two FCLs that extracts key vocabulary features from MLM predictions and maps them to label space. This approach has the advantage of promoting the efficiency of prompt-based learning without manual verbalizer manipulation or the search for optimal label words. Experimental findings not only validate the effectiveness of MAV but also highlight its self-training efficacy across all datasets, as supported by quantitative metrics and qualitative analysis.

## Limitations

Due to the instability and high sensitivity of few-shot learning to hyper-parameter selection, relatively large standard deviation can be seen in certain cases in Table 2 (Semi-supervised model with Standard Fine-tuning for TREC50, Semi-supervised model with Single Label Word for TREC50, Small-supervised model with MAV for TREC). This is mainly caused by data sampling and hyper-parameter tuning being performed separately for each seed and the small size of the validation data. Additionally, the checkpoint with the highest validation performance may not necessarily yield the best results on test dataset, resulting in significant variance. These are inherent limitations of few-shot learning. To provide clarity, we provide the performance by seed for these three cases in Appendix A.

In this study, we did not address scenarios where the class distribution of the dataset is imbalanced. It will be worth investigating a prompt-based self-training approach that can handle imbalanced datasets, which is a significant challenge in real word applications. Furthermore, we suggest exploring self-training methodologies tailored to the prompt framework that align with MLM objectives, moving beyond the application of self-training methodologies from the vision field.

## Ethics Statement

We honor the ACL Code of Ethics. All datasets employed in this study are publicly available, and

following previous research (Gao et al., 2021a), five sub-datasets were randomly sampled and utilized in the experiments. To ensure reproducibility and facilitate future research on this topic, we release all resources including datasets and code, and appropriate citations to previous research are indicated within the code.

Furthermore, our research focused on investigating news topic and sentiment classification tasks that are not anticipated to have any negative social impacts. However, when it comes to classification tasks that have the potential to raise ethical concerns regarding race and gender, it is crucial to adopt a more careful and cautious approach while analyzing the results.

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

# A   Inherent Limitations of Few-shot Learning

As mentioned in Limitations, we provide the performance by seed in the cases where relatively large standard deviation can be seen. Three cases are described in Table 6.

# B   Experimental Details

## B.1   Datasets

As elaborated in Section 4.1.1, it was essential to omit certain classes from TREC50 and GoEmotions to maintain a balanced self-training framework, considering the minimum data requirement of 80 examples per class. We aimed to ensure the integrity of training, as extremely small sample sizes for certain classes could hinder robust model learning and data consistency. Some of the eliminated classes consist of only a handful of samples, as low as 4 or 6.

Since it is not possible to make predictions for classes that are not included in the training, classes that are removed from the train and dev datasets are correspondingly excluded from the test dataset. Regarding class balance, the characteristics of test sets differ across datasets. In the case of Yahoo Answers and AG's News, the test sets are intentionally designed to maintain a balanced distribution of classes. Conversely, TREC, TREC50, and GoEmotions exhibit class-imbalanced test sets, reflective of their original nature.

## B.2   Hyper-parameter selection

**Hidden dimension of vocab extractor**   The first FCL of MAV, vocab extractor, is designed to automatically extract the vocabulary features required for classification from high-dimensional($|V|$) vocabulary logits and map them into a low-dimensional($d_{\mathrm{ve}}$) space. To identify an appropriate dimension for feature extraction, we adopted four comparison groups. As shown in Table 7, the effectiveness of MAV was greatest when the dimensionality was 256.

**Augmentation methods**   In this study, we adopted dropout and random masking as weak and strong augmentation techniques for consistency regularization, the same as SFLM (Chen et al., 2021). Unlike image data, text is characterized by its discrete nature within the language domain, where small variations can cause large semantic changes (Feng et al., 2021). Therefore, to preserve

| Method | Type | Datset | Seed 13 | Seed 21 | Seed 42 | Seed 87 | Seed 100 | Avg Acc (SD) |
|---|---|---|---|---|---|---|---|---|
| **Single** | **Semi-supervised** | **TREC50** | 89.3 | 81.2 | 86.8 | 89.6 | 60.7 | 81.5 (12.1) |
| **Standard FT** | **Semi-supervised** | **TREC50** | 84.3 | 80.2 | 84.8 | 85.0 | 62.9 | 79.4 (9.4) |
| **MAV** | **Small-supervised** | **TREC** | 63.0 | 87.6 | 71.8 | 84.0 | 79.2 | 77.1 (9.9) |

Table 6: Performance of each seed for cases where standard deviation is greater than 9. Note that Standard FT indicates Standard Fine-tuning, and Single indicates Single Label Word.

| | | TREC | TREC50 | GoEmotions |
|---|---|---|---|---|
| **hidden dimension ($d_{ve}$)** | 64 | 83.5 (4.1) | 76.8 (5.1) | 22.1 (2.4) |
| | 128 | 85.0 (3.8) | 82.5 (6.6) | 23.9 (1.1) |
| | 256 | **85.4 (3.7)** | **87.9 (5.1)** | **25.4 (2.5)** |
| | 512 | 84.7 (4.2) | 84.0 (9.3) | 25.2 (2.1) |

Table 7: Self-training performances depending on the hidden dimension of MAV.

the semantic information of unlabeled samples, it is reasonable to construct weakly augmented samples the same as the original, and let dropout in the model act as a minimal data augmentation of hidden representations (Gao et al., 2021b).

On the other hand, strong augmentation plays a significant role in performance, as emphasized by Sohn et al. (2020). Chen et al. (2021) considered only EDA techniques (*Crop, Swap, and Deletion*) as a comparison group for strong augmentation methods. However, given the diversity of augmentation techniques in the NLP field, a more rigorous comparison experiment is necessary. Thus, we employed the expanded augmentation pool provided by Kim et al. (2022). In addition to EDA, contextual augmentation and continuous augmentation are added to ensure that a sufficient variety of augmentation techniques can be considered. A total of eight augmentation techniques utilized in this study are as follows:

- **Random Mask** (Devlin et al., 2019): randomly masking tokens with a probability of 15%,

- **Word Delete** (Wei and Zou, 2019): removing each token with a probability of 20%,

- **Word Swap** (Wei and Zou, 2019): swapping the position of each token with another token with a probability of 20%,

- **Word Delete/Swap** (Wei and Zou, 2019): randomly choosing between Word Delete and Swap for each sample,

| Augmentation Methods | | TREC50 | Yahoo |
|---|---|---|---|
| Single | Random Mask (2019) (ours) | **87.9 (5.1)** | **68.1 (0.9)** |
| | Word Delete (2019) | 83.4 (4.3) | 68.0 (1.3) |
| | Word Swap (2019) | 87.2 (4.5) | 68.1 (1.0) |
| | Word Delete, Swap (2019) | 84.9 (5.3) | 67.9 (1.3) |
| | BERT-aug (2021) | 84.6 (5.9) | 67.2 (1.4) |
| | Back-translation (2020) | 85.7 (5.6) | 67.5 (0.7) |
| | R3F (2021) | 87.5 (5.6) | 67.5 (1.1) |
| | Cutoff (2020) | 84.9 (5.5) | 67.0 (1.1) |
| Random Augmentation | | 85.1 (5.9) | 67.9 (0.9) |
| Auto Augmentation (2022) | | 85.2 (3.3) | 67.7 (1.3) |

Table 8: Self-training performances depending on strong augmentation methods. Note that weak augmentation is fixed with dropout.

- **BERT-aug** (Yi et al., 2021): applying random masking with 15% probability and replacing with the result of pre-trained BERT inference,

- **Back-translation** (Xie et al., 2020): translating each sample to English-German-English,

- **R3F** (Aghajanyan et al., 2021): injecting noise sampled from a uniform distribution into word embeddings,

- **Cutoff** (Shen et al., 2020): removing from word embeddings by a factor of 0.3.

With these augmentation methods, we designed three comparison experimental settings as follows:

- **Single Augmentation**: applying one augmentation technique in the augmentation pool,

- **Random Augmentation**: randomly sampling augmentation techniques per batch (similar to FixMatch's augmentation),

- **Auto Augmentation** (Kim et al., 2022): updating the augmentation policy every iteration to select augmentation techniques that are difficult but not too semantically different from the original.

The results are shown in Table 8. Random Mask achieved the best performance, demonstrating the

| Dataset | {Class: *Single Label word*} |
| --- | --- |
| AG's News | {World: *World*}, {Sports: *Sports*}, {Business: *Business*}, {Sci/Tech: *Tech*} |
| TREC | {Abbreviation: *Expression*}, {Entity: *Entity*}, {Description and abstract concept: *Description*}, {Human being: *Human*}, {Location: *Location*}, {Numeric value: *Number*} |
| Yahoo Answers | {Society&Culture: *Society*}, {Science&Mathematics: *Science*}, {Health: *Health*}, {Education&Reference: *Education*}, {Computers&Internet: *Computer*}, {Sports: *Sports*}, {Business&Finance: *Business*}, {Entertainment&Music: *Entertainment*}, {Family&Relationships: *Relationship*}, {Politics&Government: *Politics*} |
| TREC50 | {Expression abbreviated: *shortened*}, {Animal: *animal*}, {Lasting time of something: *period*}, {Invention, book and other creative piece: *creation*}, {Disease and medicine: *medical*}, {Food: *food*}, {Other entity: *other*}, {Sport: *sport*}, {Equivalent term: *equal*}, {Definition of something: *definition*}, {Description of something: *description*}, {Manner of an action: *manner*}, {Reason: *reason*}, {Group or organization of persons: *group*}, {Individual: *individual*}, {City: *city*}, {Country: *country*}, {Other location: *location*}, {State: *state*}, {Number of something: *count*}, {Date: *date*}, {Price: *money*} |
| GoEmotions | {Optimism: *optimism*}, {Neutral: *neutral*}, {Amusement: *amusement*}, {Curiosity: *curiosity*}, {Surprise: *surprise*}, {Confusion: *confusion*}, {Nervousness: *nervous*}, {Disgust: *disgust*}, {Joy: *joy*}, {Anger: *anger*}, {Gratitude: *gratitude*}, {Sadness: *sadness*}, {Disappointment: *disappointment*}, {Desire: *desire*}, {Embarrassment: *embarrassment*}, {Remorse: *remorse*}, {Realization: *realization*}, {Excitement: *excitement*}, {Admiration: *admiration*}, {Disapproval: *disapproval*}, {Caring: *caring*}, {Fear: *fear*}, {Approval: *approval*}, {Love: *love*}, {Annoyance: *annoyance*}, {Relief: *relief*} |

Table 9: Manually selected single label words for each dataset. The label words of the TREC dataset are from LM-BFF (Gao et al., 2021a), and those of the AG's News and Yahoo Answers datasets are from PET (Schick and Schütze, 2021). For the TREC50 and GoEmotions datasets, we manually chose the label word to be a single token that closely represents the class name. Note that each label word is a single token including a space symbol (e.g., 'ĠWorld').

suitability of the strong augmentation technique adopted in this study. Prompt-based self-training naturally benefits from the random masking technique since it maintains the input form containing the [MASK] token which also used in the pre-training stage.

**Hyper-parameters for training** In this section, we describe the hyperparameter tuning process associated with the training loop. We conducted a grid search for the learning rate within the range of $\{1e-5, 5e-5\}$, the weight of the self-training loss $(\lambda_1)$ within the range of $\{0.5, 1, 2\}$, and the weight of the auxiliary MLM loss $(\lambda_2)$ within the range of $\{0.1, 1\}$. These ranges were selected based on pilot experiments conducted on the TREC and TREC50 datasets, aiming to cover a diverse set of values.

For all experiments, we trained the models for up to 200 epochs and evaluated the performance every 20 epochs. The best model was selected based on its evaluation performance. To accommodate the GPU memory limitations, gradient accumulation was employed, allowing larger effective batch sizes. The batch size for all experiments was set to 32.

### B.3 Baselines

**Single label word** For the TREC dataset, we used manual label words provided by LM-BFF (Gao et al., 2021a), and for the AG's News and Yahoo Answers datasets, which were not tested by LM-BFF, we used the manual label words provided by PET (Schick and Schütze, 2021). For the TREC50 and GoEmotions datasets, which were not tested in either of the previous studies, we adopted the class name as a label word. If the class name is not a single token, we manually selected a similar one. All label words used in this study are described in Table 9. Furthermore, we excluded the automatic search method proposed by Gao et al. (2021a) due to increasing training cost proportional to the number of classes. Additionally, we dismissed the *demonstration* technique of sampling training instances of all classes individually and concatenating them to the input sequence due to the maximum input length.

**Multi label words** AMuLaP (Wang et al., 2022a) is more efficient than other automatic verbalizers as it does not require parameter updates to explore label words. Before fine-tuning, MLM inference is

|  | TREC | TREC50 | GoEmotions |
|---|---|---|---|
| Verbalizer-free | 76.0 (6.3) | 81.7 (5.5) | 23.2 (1.1) |
| MAV (ours) | **85.4 (3.7)** | **87.9 (5.1)** | **25.4 (2.5)** |

Table 10: Self-training performances of verbalizer-free method.

performed with a small number of labeled samples to select label words with high prediction probability for each class. The logits of these tokens are then added together and considered as the logit value of the class. In this study, we selected the top 16 tokens with high prediction probability as label words for each class, applying deduplication to prevent overlap of label words across classes.

### B.4 Training details

We employed the PyTorch[7] library for training model. The overall organization of the experimental code is derived from the code implemented in previous studies[8]. Whole training process of this research was conducted using NVIDIA RTX A6000.

## C Comparison of MAV vs Verbalizer-free Approach

MAV shares similar goals with verbalizer-free methodologies as both utilize the [MASK] representation to construct soft label words. The distinction lies in MAV proceeding with MLM prediction, while most verbalizer-free methodologies rely on spatial distance from the [MASK] representation.

As mentioned in Section 2.2, existing verbalizer-free methodologies are not applicable to self-training algorithms based on pseudo-labeling and consistency regularization, which require explicit output probability distributions. Thus, a direct performance comparison with the proposed MAV is not feasible. Alternatively, we considered using the [MASK] representation fed into the classification head as a verbalizer-free baseline. The results are shown in Table 10. These findings demonstrate that leveraging all the information in the vocabulary logits produced by MLM prediction is more effective for multi-class classification than simply using the [Mask] representation.

---

[7] https://pytorch.org/
[8] https://github.com/princeton-nlp/LM-BFF,
https://github.com/MatthewCYM/SFLM