# OpenReview forum: "Boosting Prompt-Based Self-Training With Mapping-Free Automatic Verbalizer for Multi-Class Classification"
_EMNLP/2023/Conference — EMNLP 2023 Findings_

### Official Review · Reviewer_jFV1 · 2023-08-03

**Typos Grammar Style And Presentation Improvements:** 1. Section 4.4 and Table 3 are somewh…
**Soundness:** 3

**Excitement:**

4: Strong: This paper deepens the understanding of some phenomenon or lowers the barriers to an existing research direction.

**Paper Topic And Main Contributions:**

This paper focuses on a prompt-based fine-tuning on multi-class text classification, specifically where the number of classes is greater than or equal to 4. Given that creating a manual verbalizer is expensive and time-consuming and that automatically searching for optimal label words is computationally costly, the paper proposes a method called mapping-free automatic verbalizer (MAV) on top of a previous prompt-based self-training method, SFLM (Chen et al., 2021). The proposed method circumvents the aforementioned issues by appending two learnable fully-connected layers to the MLM head of RoBERTa. Experimental results demonstrate the effectiveness of the proposed method in a few-shot setting on five datasets, where the number of training samples K is set to 16, and the number of classes ranges from 4 to 26. The paper also provides a number of analyses to support the superiority of the proposed method over the comparison methods.

Updated: I have read the author rebuttal and keep the scores unchanged. While the rebuttal makes sense, current experimental results with only RoBERTa-base and K=16 raise the question about the generalizability of the proposed method with other language models with different sizes as well as different values of K.

**Questions For The Authors:**

A. Why is there no evaluation on larger LMs such as RoBERTa-large (355M parameters) as the previous work (Gao et al. 2021a) did? Fine-tuning becomes much more expensive as the size of LMs grows and few-shot fine-tuning plays a critical role especially when fine-tuning larger LMs. Since the paper experiments in a few-shot setting, an evaluation on larger LMs is recommended.

B. How sensitive is the proposed method to the value of K, the number of examples per class? It would be interesting to compare it with other baselines.

C. Why is there no evaluation using the entire dataset? It is assumed that the proposed method can be fine-tuned on the entire dataset without filtering. This experiment is needed to evaluate the applicability of the proposed method to any multi-class datasets.

D. How is the test set for each dataset created? Given that the datasets are pre-processed based on the number of classes, as shown in Table 1, it is supposed that the test set is also pre-processed. Furthermore, while the train and dev sets are class-balanced, is the test set the same?

E. Why is the fine-tuning process so long? In general, the number of epochs is set to a small value, such as 3, 5, or 10, in fine-tuning (Gao et al. 2021a; Liu et al. 2019). However, the paper uses 200 for the number of epochs (Line 977), which seems to be a bit larger value.

**Reasons To Accept:**

1. The proposed method is well-motivated and addresses issues of manual verbalizer creation that requires domain knowledge and an increasing cost for automatically searching label words correlating with the number of target classes.
2. The paper is well-structured and easy to follow. Sufficient descriptions of the experimental setup are provided, which ensures reproducibility.

**Reasons To Reject:**

1. The paper conducts the experiments only with a single pre-trained language model, RoBERTa-base, which is also the smallest one (125M parameters) in size among RoBERTa variants, and a single few-shot setting (K=16), both of which narrow the generalizability of the proposed method.
2. The paper argues that the proposed method can be applied to any multi-class datasets, irrespective of the number of classes (Lines 109-110), while the experiments are conducted on filtered datasets, as shown in Table 1, where the minor classes are removed (e.g., the number of classes decreased from 50 to 22 in Trec50) to meet the requirement regarding the number of examples per class to fine-tune the proposed method. This may contradict the argument and limit the applicability of the proposed method to any multi-class datasets.

**Reproducibility:**

4: Could mostly reproduce the results, but there may be some variation because of sample variance or minor variations in their interpretation of the protocol or method.

**Reviewer Confidence:**

4: Quite sure. I tried to check the important points carefully. It's unlikely, though conceivable, that I missed something that should affect my ratings.

---

> ### Author Rebuttal · Authors · 2023-08-29
>
> We extend our gratitude for taking the time to provide us with invaluable comments and inquiries regarding our paper. Your acknowledgment of our method as being well-motivated, and your recognition of the paper's clear structure and comprehensive descriptions, are appreciated. The detailed responses to your main concerns and questions are as below:
>
> **Why is there no evaluation on larger LMs such as RoBERTa-large (355M parameters) as the previous work (Gao et al. 2021a) did? (*Question A*)**
>
> We would like to clarify our rationale for the model selection in our study. Prior investigations in the realm of prompt-based fine-tuning have employed a range of pre-training language models. Notably, these include Gao et al. (2021a) and Hu et al. (2022) with RoBERTa-large, Chen et al. (2021) with RoBERTa-base, and Zhao and Yao (2022) with DistilRoBERTa. In our study, we deliberately opted for RoBERTa-base to facilitate optimal comparisons with the most closely related prompt-based self-training literature, the SFLM approach of  Chen et al. (2021). While earlier research in prompt-based fine-tuning with only limited labeled data might have sought to harness the advantages of larger models to enhance few-shot fine-tuning efficacy, our emphasis was on the self-training process with additional unlabeled data rather than the scale of the model.
>
> **How sensitive is the proposed method to the value of K, the number of examples per class? (*Question B*)**
>
> Drawing on insights from prior prompt-based fine-tuning studies (Gao et al., 2021a; Chen et al., 2021), it's evident that employing a smaller K (i.e., fewer labeled samples) can yield significant performance advantages over classical fine-tuning. While these studies have observed performance variations with varying K values, their experimental setup is conducive to constructing datasets for comparative analysis due to their focus on binary or 3-class classification scenarios.
>
> However, our study faces a unique challenge in this context. The limited amount of available data impedes our ability to conduct experiments with varying K values. A crucial constraint emerges from the necessity to omit classes with insufficient data to achieve a balanced setting, thereby hindering the establishment of a fair comparison framework.
>
> **Why is there no evaluation using the entire dataset? (*Question C*)**
>
> As elaborated in Section 4.1.1, the decision to exclude certain classes from datasets like TREC50 and GoEmotions was rooted in the necessity to maintain a balanced self-training framework, given the minimum data requirement of 80 examples per class. This choice aimed to ensure the integrity of training, as extremely small sample sizes for certain classes could hinder robust model learning and data consistency. Some of the eliminated classes consist of only a handful of samples, as low as 4 or 6.
>
> Addressing such imbalanced setting lies beyond our current scope. We acknowledge it in the Limitations section and propose it as potential future work. To openly handle this concern, we plan to add appendix with a table detailing the classes that were excluded from each dataset, enhancing clarity for readers.
>
> **How is the test set for each dataset created? While the train and dev sets are class-balanced, is the test set the same? (*Question D*)**
>
> Since it is not possible to make predictions for classes that are not included in the training, classes that are removed from the train and dev datasets are correspondingly excluded from the test dataset.
>
> Regarding class balance, the characteristics of test sets differ across datasets. In the case of Yahoo Answers and AGNews, the test sets are intentionally designed to maintain a balanced distribution of classes. Conversely, TREC, TREC50, and GoEmotions exhibit class-imbalanced test sets, reflective of their original nature. We will supplement the information about the test datasets in order to enhance the overall clarity of our work.
>
> **Why is the fine-tuning process so long (200 epochs)? (*Question E*)**
>
> We selected the number of training epochs based on careful consideration of existing literature. Gao et al. (2021a) and Chen et al. (2021) employ a maximum step of 1000, with validation conducted every 100 steps. Taking into account details, where the number of labeled samples per class (k) is 16 and the maximum batch size used for hyper-parameter tuning is 8, the training epoch range is estimated to be around 160 to 250 epochs for binary or 3-class classification.
>
> Given the increase in the number of classes in our dataset compared to previous studies, we opted not to simply replicate the same max step value. Instead, we set 200 epochs as the maximum number of training iterations. We also set the validation interval to 20 epochs to ensure equal intervals.
>
> It's important to highlight that prompt-based fine-tuning diverges from regular fine-tuning in that it doesn't operate with the full dataset, contributing to the relatively extended process duration.
>
> **More details for Section 4.4 and Table 3**
>
> The experiment in Section 4.4 was designed to compare and analyze the impact of freezing the MLM head, a key feature of how MAV works. This freezing mechanism aims to capitalize on the pre-trained vocabulary representation inherent in the decoder of the MLM head. The three settings shown in Table 3 are: updating all parameters, freezing only the MLM head, and freezing the RoBERTa encoder. Basically, we compared the performance of prompt-based self-training based on the MAV structure with different parameter update scenarios while following the overall self-training framework in Figure 2.
>
> As evidenced in Table 3, the results of freezing the MLM head's parameters closely align with those achieved through full fine-tuning. At the same time, it significantly outperforms the scenario where the RoBERTa encoder is frozen. This suggests that the MLM head is sufficiently adept at generating MLM predictions that carry valuable information about the downstream data even without further tuning.
>
> In response to your feedback, we will enhance the clarity of Section 4.4 by elaborating on the experimental setting surrounding Table 3.

---

### Official Review · Reviewer_iK6f · 2023-08-03

**Soundness:** 3

**Excitement:**

3: Ambivalent: It has merits (e.g., it reports state-of-the-art results, the idea is nice), but there are key weaknesses (e.g., it describes incremental work), and it can significantly benefit from another round of revision. However, I won't object to accepting it if my co-reviewers champion it.

**Paper Topic And Main Contributions:**

This paper deals with the problem of few-shot multi class classification, specifically in the context of performing the classification using the masked language modeling capabilities of an encoder language model. The authors propose a new approach for a learnable verbalizer that converts the masked token predictions of the model to class predictions. The approach consists of adding two layers on top of the MLM head for learning the classification task with few-shot labeled data, while keeping the pre-trained MLM head frozen. The authors combine their verbalizer approach with existing approaches for self-training in the context of prompt-based fine-tuning, and compare this with baseline verbalizer methods over 5 multi class datasets. The main contribution is in proposing the new approach for a learnable verbalizer for few-shot classification, which importantly does not require manually specifying class label names; the authors show that their approach outperforms existing verbalizer approaches for few-shot prompt-based fine-tuning, especially where self-training is used.

**Questions For The Authors:**

A. In §4.5, what is the justification for calculating SHAP values only for the correct predictions?

**Reasons To Accept:**

- Overall, the methodology and the comparisons to different baselines look sound, and code for reproducing the results is provided.
- The authors demonstrate strong few-shot performance for the proposed approach - that does not require any manual label names - even outperforming baselines that do require a manual label. The performance numbers are also backed by analyses that show the approach induces better separation of mask representations (§4.3) and results in logical verbalizer behavior (§4.5).
- The approach of adding layers on top of the MLM head is interesting. In a sense this is a combination between classic fine-tuning - where we simply train a new classification head that converts model outputs to class labels, and prompt-based fine-tuning - where we rely on the knowledge of the MLM head to map to a set of class label names. So here we still rely on the MLM knowledge to some extent, but nevertheless need to learn the task-specific mapping on top of that.


**Reasons To Reject:**

- In my view there is a certain lack of focus in the overall narrative of the paper, in terms of more precisely describing what is the problem the paper aims to solve and what is the unique contribution of this work. Specifically, the problem of implementing a good verbalizer for (multi-class) classification is a relatively general one, whereas the question of the interaction of the verbalizer with self-training is more specific. There is no problem with focusing more on the latter, except I did not find in the paper either a very clear motivation for this — i.e., why would self-training specifically suffer more from a suboptimal verbalizer — or an interesting analysis of this point — i.e, showing how something about the way MAV works explains why it benefits self-training specifically.
- I found the description of the experimental setting and baselines a bit difficult to follow, in that it is not sufficiently clear to the reader what is kept constant and what differs between the settings reported in Table 2. For instance, if my understanding is correct, all the rows marked with "Semi-supervised" use the loss formulations as described in §3.2; this is not sufficiently clear from the text.
 On a related note, I think the structure of results in Table 2 can be improved, as it makes it hard for the reader to see the comparison between different approaches over the same setting (as these rows are not adjacent to each other). For example, what is arguably the core result in this paper - that MAV outperforms the closest prior work of SFLM, gets a bit lost in the table. Put another way, if my problem is text classification and what I have are few-shot labeled examples, what I would want to know is which method works best in this scenario; if I also have unlabeled data and can do ST, I want to know what works best for that; and if I have a lot of labeled examples, then again I would want to know which approach to choose, and the results can be presented in a way that more clearly conveys this information. In contrast, knowing what is the "benefit ratio" for each method is less useful, assuming I am interested in dealing with a specific scenario (e.g. few-shot) and want to reach a better absolute performance on a downstream task.
- Comparing freezing the MLM head to freezing the entire encoder (§4.4) is a bit odd, as it is not the same scale of trainable parameters. In any case, while it is clear that freezing the MLM head does not hurt performance, concluding from this that it is "able to produce MLM predictions that are sufficiently informative about the downstream data on its own" (l. 475) is not really substantiated by Table 3.


**Reproducibility:**

5: Could easily reproduce the results.

**Reviewer Confidence:**

4: Quite sure. I tried to check the important points carefully. It's unlikely, though conceivable, that I missed something that should affect my ratings.

**Typos Grammar Style And Presentation Improvements:**

l. 130: answer to a critical research -> answer a critical research
l. 142: of manual approach -> of the manual approach
l. 237: MAV autonomously identify -> MAV autonomously identifies

---

> ### Author Rebuttal · Authors · 2023-08-29
>
> Thank you for valuable observations in enhancing the clarity and impact of our paper. We appreciate your interest in our methodology and the robustness of our comparisons, as well as your acknowledgment of the code's availability for result reproduction. We are also pleased that you recognize that our performance numbers are supported by the analyses in §4.3 and §4.5. We address your main concerns as below:
>
> **Why MAV benefits self-training?**
>
> We value your insightful perspective on the overall narrative of the paper and its focus. We acknowledge the importance of clarifying the motivation behind our emphasis on the interaction of the verbalizer with self-training.
>
> Our paper focuses on the prompt-based self-training task, which involves prompt-based fine-tuning and semi-supervised learning (especially, self-training based on pseudo-labeling and consistency regularization such as FixMatch). In this context, an effective verbalizer must address two aspects: (1) identifying suitable label words for prompt-based fine-tuning and (2) generating explicit probability distributions for self-training. Current manual/automatic verbalizers derive predictive distribution through label word predictions but are inefficient due to the cost of label word selection. The verbalizer-free approach mentioned in §2.1 lacks the ability to derive probability distributions due to distance-based learning.  Our contribution, MAV, addresses these issues by eliminating label word selection overhead in prompt-based fine-tuning and generating explicit probability distributions for self-training. This design bridges the gap between verbalization and self-training, addressing the challenges of both aspects in a unified manner.
>
> **More details for experimental settings and Table2**
>
> As you clarified, "Small-supervised" and "Full-supervised" in Table 2 correspond to the models trained with the supervised loss outlined in §3.2, while "Semi-supervised" represents the model trained with both self-training loss and auxiliary MLM loss in addition to supervised loss. This distinction will be explicitly added to the text and table captions for clarity.
>
> Furthermore, within the context of Table 2, the amalgamation of "Small," "Semi," "Full," and the "Benefit ratio" metrics serves a crucial purpose for grasping the efficacy of self-training. "Small-supervised" showcases how "Semi-supervised" benefits from incorporating unlabeled data with limited labeled data, while "Full-supervised" represents the upper performance limit in the semi-supervised context. Moreover, the "benefit ratio" metric provides a quantitative means to comprehend the interplay between these configurations and the effect of self-training.
>
> Recognizing the need for enhanced clarity, we will consider reorganizing Table 2, separating "Small," "Semi," and "Full" into distinct subtables for intuitive comparison.
>
> **Effect of freezing the MLM head**
>
> We appreciate your comment regarding the comparison between freezing the MLM head and freezing the entire encoder in §4.4. We understand your point about the difference in the scale of trainable parameters between these two scenarios, which indeed presents a nuanced aspect to consider.
>
> Your observation about the conclusion drawn from Table 3 is also well-taken. Our objective in freezing the MLM head is to generate informative predictions that are not biased by the small amount of labeled data during the fine-tuning process. Simultaneously, we harness the pre-trained knowledge embedded within the model. Nonetheless, we acknowledge that while the results show that freezing the MLM head doesn't negatively impact performance, the direct inference that it can produce sufficiently informative MLM predictions for downstream data requires further substantiation. In our revised manuscript, we will aim to address the comparison more judiciously by emphasizing the distinction in parameter scales between the scenarios.
>
> **In §4.5, what is the justification for calculating SHAP values only for the correct predictions? (*Question A*)**
>
> The reason for this approach is rooted in the nature and intent of §4.5, which focuses on post-hoc analysis rather than performance evaluation. The primary aim is to gain insights into the mechanisms by which MAV identifies relevant vocabulary features for classification, rather than assessing overall model performance.
>
> Given this context, we deemed it appropriate to calculate SHAP values solely using data with correct predictions. Excluding incorrect answer data minimizes the potential noise introduced by instances where the model's prediction might not align with the actual class associations.

---

### Official Review · Reviewer_jSyo · 2023-08-15

**Typos Grammar Style And Presentation Improvements:** N/A
**Soundness:** 4

**Excitement:**

4: Strong: This paper deepens the understanding of some phenomenon or lowers the barriers to an existing research direction.

**Missing References:**

N/A

**Paper Topic And Main Contributions:**

The manuscript proposed a multi-class prompt-based self-training framework with an automatic verbalizer. By utilizing the trained verbalizer, the framework provides a learnable and mapping-free form for self-training. Experiment results demonstrate the effectiveness of the proposed framework by comparison with the latest methods.

**Questions For The Authors:**

a, More details for the "Multi Label Words Mapping" step in Fig. 1 is recommended to be provided, as there exit various solutions for this task;
b, What is the masking strategy difference between the "Random Masking" operation for "Auxiliary MLM loss" vs. the "Strong-Aug (Random Masking)" operation for "Self-training loss"?
c, In lines 229-230, is the low-dimensional vocabulary d_ve learned during the training or a pre-fixed set?

**Reasons To Accept:**

Prompt-based self-training is a promising framework for FSL tasks in NLP. Extending the current works into a multi-class scheme by adopting the automatic verbalizer is an algorithmically feasible solution.

Empirical results support the author's premise well.

**Reasons To Reject:**

Considering the multi-stage training of the MLM head and the multiple loss functions involved in this framework, re-implementation of the work and reproducing the results would be quite challenging. The paper did not provide any source code although it uses public datasets. B.3 might not be as easy as it describes.

**Reproducibility:**

2: Would be hard pressed to reproduce the results. The contribution depends on data that are simply not available outside the author's institution or consortium; not enough details are provided.

**Reviewer Confidence:**

2: Willing to defend my evaluation, but it is fairly likely that I missed some details, didn't understand some central points, or can't be sure about the novelty of the work.

---

> ### Author Rebuttal · Authors · 2023-08-29
>
> Thank you for the valuable comments. It's encouraging to hear that you view our work as both promising and feasible for tackling multi-class classification challenges, and that the empirical results have effectively supported our claims. We have carefully considered your main concerns and have addressed them in the following manner:
>
> **Source code for this paper**
>
> We would like to highlight that we have diligently prepared the source code for reproduction purposes. This comprehensive code, available as Supplementary Materials, is accompanied by detailed documentation(`readme.md`) outlining the experimental procedure and hyper-parameter specifications. We believe that this effort will address any reservations regarding reproducibility you might have.
>
> **More details for Multi Label Words Mapping (*Question A*)**
>
> We would like to clarify more details for Multi Label Words Mapping.
>
> Label word mapping is a technique that specifies one or several label words corresponding to each class before fine-tuning. Subsequently, only the predicted values of these label words are extracted from the MLM predictive distribution, constructing the final probability distribution for multi-class classification. If more than one label words are selected from the word dictionary, which has a vocab size of 50265 for RoBERTa-base, it becomes “Multi Label Words Mapping”. The selection of label words can be performed manually or via automatic methods.
>
> Fig. 1 (b) visualize the process of Multi Label Words Mapping in multi-class classification task with 6 classes. A set of multiple label words corresponding to each class is defined (e.g., V_Location). The sum of MLM predictions for label words within this set (e.g., *Location, City, Place,* ...) forms the predicted probability for the "Location" class. In this paper, we used AMuLaP (Wang et al., 2022a), an automatic label word searching approach, as a baseline for “Multi Label Words Mappping”. AMuLaP involves zero-shot inference before fine-tuning with only a small amount of labeled data (about 16 examples per class) to identify top-k (we set the value 16) words with substantial MLM prediction probabilities. This methodology exhibits strength in binary classification tasks where the distinction between classes is clear. However, as shown in Table 4, under multi-class dataset where the distinction between classes is relatively vague, tokens unrelated to “Location” class such as *Poll* and *Related* are selected as label words, which leads to the poor performance of Multi Label Words Mapping in Table 2.
>
> **"Auxiliary MLM loss" vs. "Strong-Aug (Random Masking)" (*Question B*)**
>
> We would like to highlight that the "Self-training loss" and the "Auxiliary MLM loss" serve different purposes and utilize distinct masking outcomes of unlabeled samples.
>
> The self-training process endeavors to ensure that the predicted distribution of the strongly augmented samples aligns closely with the pseudo-labels derived from the weakly augmented samples. Hence, in this scenario, random masking exclusively functions as an strong augmentation technique. As depicted in Fig. 2 `Strong-Aug (Random Masking)`, only the masked token within the template, denoted in red, traverse through PLM, MLM head and MAV, generating final predictive distribution used in the computation of the self-training loss.
>
> On the other hand, for "Auxiliary MLM loss", the unlabeled input is independently masked, separate from strong augmentation. The random Masking applied in this context aims to enhance the model's ability to learn diverse contextual relationships by masking tokens in the unlabeled samples. This allows pre-trained language model to preserve the ability to perform MLM during prompt-based fine-tuning process. Therefore, in this scenario, the MLM loss derived from the newly masked token, denoted in red, serves as an auxiliary loss (`Random Masking` at the bottom of Figure 2).
>
> **The dimension of vocab extractor, d_ve (*Question C*)**
>
> During the fine-tuning process, all parameters of the two fully connected layers comprising  MAV are learned. The hidden dimension (d_ve) of the first layer, named vocab extractor, is a pre-fixed value (256) determined by hyper-parameter tuning involving four comparison groups ({64, 128, 256, 512}). Details for the tuning can be found in Appendix B.1 and Table 7.

---

### Meta-Review · Area_Chair_Rgoc · 2023-09-20

**Recommendation:** 2

**Metareview:**

This paper presents a method for multi-class few-shot and semi-supervised classification with masked language models. Previous approaches use manually defined verbalizers that map from class labels to tokens in the MLM’s vocabulary, but this does not scale well to datasets with more classes. The method instead adds a classification head on top of the distribution over tokens outputted by the MLM head. Experiments are on text classification tasks and often combine the method with self-training on unlabeled examples. Reviewers found the method appropriate and appreciated the strong empirical results and additional analysis. As pointed out by reviewer jFV1, the experiments are not the most thorough: only covering one pretrained model and a fixed number of labels per class. It also isn’t justified why going through the vocabulary is necessary at all as opposed to applying the classification head to the [MASK] hidden state directly. Additionally, the experiments filtered out classes from the datasets with very few examples per class, which I find strange because this is precisely the setting where we want good few-shot learning. Lastly, as reviewer iK6f suggests, the paper could be clearer about the problem being solved: while the paper largely discusses few-shot learning, the results for the pure few-shot setting (with no unlabeled data) are often worse than the single word label baseline. Instead, the method really shines in the semi-supervised setting, but there isn’t much analysis on why it benefits more from self-training than other approaches. To summarize, the method is well-motivated and produces some strong results, but the paper could be clearer about when and why the method works.

---

### Decision · Program_Chairs · 2023-10-07

**Decision:**

Accept-Findings

**Comment:**

This paper presents a method for multi-class few-shot and semi-supervised classification with masked language models. Previous approaches use manually defined verbalizers that map from class labels to tokens in the MLM’s vocabulary, but this does not scale well to datasets with more classes. The method instead adds a classification head on top of the distribution over tokens outputted by the MLM head. Experiments are on text classification tasks and often combine the method with self-training on unlabeled examples. Reviewers found the method appropriate and appreciated the strong empirical results and additional analysis. As pointed out by reviewer jFV1, the experiments are not the most thorough: only covering one pretrained model and a fixed number of labels per class. It also isn’t justified why going through the vocabulary is necessary at all as opposed to applying the classification head to the [MASK] hidden state directly. Additionally, the experiments filtered out classes from the datasets with very few examples per class, which I find strange because this is precisely the setting where we want good few-shot learning. Lastly, as reviewer iK6f suggests, the paper could be clearer about the problem being solved: while the paper largely discusses few-shot learning, the results for the pure few-shot setting (with no unlabeled data) are often worse than the single word label baseline. Instead, the method really shines in the semi-supervised setting, but there isn’t much analysis on why it benefits more from self-training than other approaches. To summarize, the method is well-motivated and produces some strong results, but the paper could be clearer about when and why the method works.